# A Pilot Study to Assess the Safety and Efficacy of Umbilical Cord Blood-Derived Mesenchymal Stromal Cells for the Treatment of Synovitis in Horses

**DOI:** 10.3390/ani14233406

**Published:** 2024-11-26

**Authors:** Kathryn Seabaugh, Sangeeta Rao, Judith B. Koenig, Lynn Pezzanite, Steven Dow, Thomas G. Koch, Keith A. Russell, Sahar Mehrpouyan, A. Hamed Alizadeh, Laurie R. Goodrich

**Affiliations:** 1Orthopaedic Research Center, Translational Medicine Institute, Colorado State University, Fort Collins, CO 80523, USA; lynn.pezzanite@colostate.edu (L.P.); steven.dow@colostate.edu (S.D.); 2Department of Clinical Sciences, College of Veterinary Medicine and Biomedical Sciences, Colorado State University, Fort Collins, CO 80523, USA; sangeeta.rao@colostate.edu; 3Department of Clinical Studies, Ontario Veterinary College, University of Guelph, Guelph, ON N1G2W1, Canada; jkoenig@uoguelph.ca; 4eQcell Inc., University of Guelph, 50 Stone Road East, Guelph, ON N1G 2W1, Canada; tkoch@eqcell.com (T.G.K.); krussell@eqcell.com (K.A.R.); smehrpouyan@eqcell.com (S.M.); halizadeh@eqcell.com (A.H.A.)

**Keywords:** synovitis, equine umbilical cord blood-derived mesenchymal stromal cells, equine lameness

## Abstract

Synovitis frequently affects joints in horses and can result in lameness. Synovitis results in joint capsule distention and inflammation of the joint environment, which can lead to more serious complications, such as osteoarthritis, if not addressed early. Researchers are searching for a treatment to interrupt this progression and address the synovitis prior to osteoarthritis developing. We present a study that assesses the safety and efficacy of a single administration of 10 million cryopreserved allogeneic mesenchymal stromal cells (MSC) that were either activated (aMSC) or non-activated (naMSC) for the treatment of synovitis. The front or hind fetlock or a single joint of the carpus in 24 client-owned horses were split into two groups. Both MSC treatments resulted in positive clinical outcomes. When evaluating from Day 0 to Day 42, naMSC-treated horses improved by 0.79 lameness grades and aMSC-treated horses improved by 1.25 lameness grades. There was no significant difference between the two treatments for any outcome parameter. Owners were contacted at 12 and 18 weeks after treatment. All fifteen owners who returned questionnaires at 12 weeks were satisfied with their horses’ response to treatment. Eighteen of the twenty-three questionnaires that were returned at Week 18 indicated that the owners were satisfied with their horses’ response to treatment.

## 1. Introduction

Osteoarthritis (OA) poses one of the most devasting threats to the joints of horses, regardless of discipline. It continues to be one of the most significant challenges for researchers to treat and prevent. Synovial inflammation is a major factor in OA and can originate from acute traumatic injury to the joint, repetitive trauma, or infection. Synovitis triggers a pro-inflammatory response and affects the entire joint environment, which can cause damage to the articular cartilage extracellular matrix, or even death and the apoptosis of chondrocytes [1]. This disruption to the joint ecosystem dramatically alters the composition of synovial fluid and leads to decreased lubrication of the joint. Although the lubricative ability of the synovial fluid varies widely between each horse, repetitive movement within the joint paired with any degree of lessened lubricative properties of the synovial fluid causes changes to weight-bearing surfaces and can play a substantial role in contributing to the progression of osteoarthritis [1].

Orthobiologics have become a popular and efficacious treatment for synovitis and osteoarthritis. Multiple orthobiologic options exist, and choosing the ideal orthobiologic is a challenge for equine veterinarians. Mesenchymal stromal cells (MSC) have been a popular orthobiologic option for intra-articular treatment, but using autologous MSC requires harvesting and expanding these cells, which requires time and morbidity to the horse and higher expense to the client versus an “off-the-shelf” option. MSCs can originate from a variety of tissue sources, including, but not limited to, bone marrow, adipose tissue, umbilical cord, and dental pulp [2]. In recent years, allogeneic MSC options have become more readily available. These products offer an alternative to an autologous option with the ability to control the quality and quantity of MSCs, resulting in a more uniform cellular therapy product [3]. Further research has found that MSCs can be activated to increase their immunomodulatory properties [4,5] which could enhance their anti-inflammatory capabilities.

The current study evaluates the efficacy of allogeneic equine umbilical cord blood-derived mesenchymal stromal cells (MSC) for the treatment of synovitis in horses. The authors hypothesized that intra-articular treatment with either activated or non-activated MSCs would significantly improve the lameness scores of treated horses and that treatment would result in at least one grade of improvement in lameness based on the AAEP lameness scale [6].

## 2. Materials and Methods

### 2.1. Horses

Horses were eligible for this study if they met specific inclusion criteria. Horses had to have mild to moderate lameness (1–3/5 AAEP lameness scale) [6]. The lameness had to be localized (at least 70% improvement) to the metacarpophalangeal, metatarsophalangeal, radial carpal, or middle carpal joint with intra-articular anesthesia. Lastly, at least one of the following clinical signs needed to be present: joint effusion, heat, reduced range of motion of the target joint or positive response to flexion, and/or radiographic evidence of degenerative joint disease (enthesophyte, osteophyte, sclerosis, bone proliferation).

Horses were excluded if they had received any previous MSC treatments, had an AAEP lameness score of <1 or ≥4 out of 5, or had radiographic changes consistent with severe osteoarthritis (narrowed joint space reducing ≥1/3 normal joint space, significant osteophytosis, or evidence of significant subchondral bone damage), as judged by a board-certified radiologist.

The pre-admission examination was performed by a veterinarian associated with this study or a referring veterinarian. Once a horse was deemed eligible for this study, they were referred to one of three veterinary practices. Horses must have presented to one of these study hospitals within 2 weeks but no sooner than 24 h after performing the pre-admission examination. All clients with horses that met the inclusion criteria were asked to sign an informed consent release (Appendix A).

### 2.2. Umbilical Cord Blood-Derived Mesenchymal Stromal Cells

The MSCs were harvested from donor broodmares of various ages and who were either thoroughbreds or standardbreds. All broodmares appeared healthy at the time of foaling. Only MSCs from cord blood samples with broodmares with normal complete blood counts and negative Coggins test were used. The MSCs were submitted to the Animal Health Lab (University of Guelph, ON) for antigen PCR testing and were negative for equine pegivirus, equine parvovirus, hepacivirus, Theilders disease-associated virus, West Nile virus, equine eastern encephalitis virus, equine herpes virus-1, and equine viral arteritis virus. The clinical samples were also tested for sterility by bacterial culture and endotoxin testing at the time of cryopreservation. The MSCs were isolated using the published methods from the Koch lab at the University of Guelph [7]. Cells were expanded for 4 passages. At the end of the 4 passages, the cells were characterized for their surface marker expression and their ability to suppress lymphocyte proliferation in vitro (potency assay). Only cells with a proven ability to suppress lymphocyte proliferation in vitro were released for clinical use. A proprietary activation method was used to activate the MSCs. Final cell samples were frozen in cryomedia consisting of DMEM with 10% DMSO at a concentration of 10 million cells per milliliter.

### 2.3. Clinical Evaluations

Horses were evaluated clinically at 4 timepoints: Day 0, Day 1, Day 21, and Day 42. Baseline data were acquired, and treatment was performed on Day 0. After the horses arrived at the clinic, their heart rates, respiratory rates, and rectal temperatures were recorded; signs of pre-existing inflammation were recorded through the palpation of the distal limbs and proposed treatment site. For the purpose of clarity, “subjective” refers to the grades assigned by the veterinarian for lameness, effusion, range of motion, and response to flexion, and “objective” refers to the score assigned to the horse by the inertial sensor-based lameness system (Lameness Locator^®^ 20/20 by Equinosis^®^ (v. 5.0.8395.14460), Columbia, MO, USA). The manner in which the objective scores were analyzed is described in detail in the statistical analysis section. A subjective effusion score was assigned for the target joint (Appendix A). A subjective lameness evaluation was performed, and a score was assigned (Appendix A) on a straight line and in a circle in each direction. The surface on which they were evaluated was recorded. An objective lameness score was obtained concurrently using the inertial sensor-based system. Flexion tests were performed on target joints, during which the range of motion of the joint was assessed and graded (Appendix A) and the response to flexion was assessed and graded (Appendix A).

On Day 1, 24 h after injection, the joint was assessed. Effusion was assessed and graded. Heart rate, respiratory rate, and rectal temperature were recorded. The horse was assessed for lameness prior to discharge from the clinic. This consisted of only a subjective and objective lameness grade on a straight line and in a circle in each direction. Day 1’s assessment did not include a flexion score or range of motion assessment. The assessment on Day 1 was performed by the same clinician as on Day 0. The horse was then discharged to the owner following the examination on Day 1.

On Day 21 and Day 42 post-injection, horses were subjected to follow-up lameness examinations. After arrival at the veterinary clinic, a general physical examination was performed. Heart rate, respiratory rate, and rectal temperature were recorded. The treated joint was also assessed for any abnormalities. Similar to Day 0, a lameness evaluation was performed, which again consisted of subjective and objective lameness assessments on a straight line and in circles in each direction. Flexion tests and range of motion assessments were performed on target joints. Joint effusion was graded.

### 2.4. Treatments

Treatments were assigned using a random number generator (List Randomizer https://www.random.org/lists/, accessed on 12 May 2023), whereby 12 non-activated MSC and 12 activated MSC treatments were placed in random order, and then horses were designated to a treatment group based on their order of enrollment. Horses received a single injection of either a red vial or a green vial. The vials were color-coded for blinding purposes. Upon completion of this study, the code was revealed that the red vial contained 10 million activated (aMSC) allogeneic equine umbilical cord blood-derived mesenchymal stromal cells (MSCs) and the green vial contained 10 million non-activated (naMSC) allogeneic MSCs. The product was supplied cryopreserved and stored in liquid nitrogen until required and injected immediately upon thawing. MSCs were administered via intra-articular injection. The treating veterinarians did not know if the horses were receiving aMSCs or naMSCs.

After acquiring baseline data on Day 0, horses were sedated and administered a single dose of flunixin meglumine (500 mg, IV, 0.83–1.1 mg/kg). After the horse was sedated, the injection site was aseptically prepared for 5 min, followed by a final sterile preparation. At the time of articular injection, a 20 G needle was inserted into the affected joint, and synovial fluid was collected, if possible, for a routine synovial fluid cell count and total protein count to determine baseline joint inflammation. The fluid was collected into an ethylenediamine tetra-acetic acid (EDTA, purple top) tube and submitted to a diagnostic lab for fluid analysis. With the needle in place, either 10 million aMSCs or 10 million naMSCs were injected. A sterile bandage was placed over the injection site. This injection only happened once during this study. The treatment was then complete, and animals were monitored post-injection until they recovered from sedation and stayed overnight at the participating clinic.

Detailed exercise and rehabilitation instructions for the initial 3 weeks following the treatment were supplied to the client based on Day 0 examination findings (Appendix A). The exercise and rehabilitation instructions were then modified for Weeks 3–6 based on the Day 21 exam findings.

### 2.5. Owner Questionnaires

A questionnaire (Appendix A) was provided to the owners 12 and 18 weeks after treatment to determine the level of work to which the horse had returned (not returned to work, returned to a lower level of work, the same level of work, or a higher level of work), if any adverse reactions were noted at home (yes/no), and if the owner was satisfied with the treatment (yes/no). There was also space available for the owner to provide additional comments. Owner questionnaires were emailed via an online survey platform (Survey Monkey, https://www.surveymonkey.com/, accessed on 7 September 2023). The questionnaire was sent out 12 and 18 weeks following the treatment injection to evaluate how the horses were returning to work and if any adverse events had occurred following any lameness time points. Lastly, owners were asked if they were satisfied with the treatment. Owners were offered an incentive of USD 500 for participation in the clinical trial. The study protocol required that owners complete both questionnaires to receive the incentive. If the questionnaires were not completed in a timely fashion the authors used emails and phone calls to accomplish questionnaire completion.

### 2.6. Statistical Analysis

The continuous data were described using means and evaluated for normality. If normality was not met, the data were described using medians and transformed into a log scale. A linear mixed model analysis was performed on the data to compare treatment by day. The data were adjusted for Horse ID as a random effect in the model. Adjusted *p*-values were reported. The categorical data were described using counts and analyzed using Fisher’s exact test when the counts of any of the categories were <5. SAS v9.4 (SAS Institute Inc., Cary, NC, USA) was used for all statistical analyses. A *p*-value of 0.05 was used as the criterion to determine statistical significance.

Summary statistics for data that met normality assumptions are listed as mean (±standard deviation). Summary statistics for data that did not meet normality assumptions are listed as median (interquartile range).

Lameness amplitude during all activities (straight, lunge left, lunge right) was quantified using the Equinosis Q with Lameness Locator, a body-mounted inertial sensor system designed to measure head and pelvic trajectory asymmetry for forelimb and hind limb lameness. Forelimb lameness amplitude was expressed as the vector sum (VS) of the maximum head height difference (HDmax) and the minimum head height difference (HDmin) between the right and left forelimb steps of a stride (VS = √(HDmax^2^ + HDmin^2^)). Positive HDmin values indicate right forelimb lameness, while negative values indicate left forelimb lameness, with VS assigned the same sign as HDmin. Hind limb lameness amplitude was measured using the maximum pelvic height difference (PDmax) and minimum pelvic height difference (PDmin) between the right and left hind limb steps of a stride. Positive values of PDmin and PDmax signify right hind limb lameness, while negative values indicate left hind limb lameness. PDmin reflects the impact component (reduced downward pelvic excursion), and PDmax reflects the push-off component (reduced upward pelvic excursion) of hind limb lameness. Prior to calculating changes in lameness amplitude between treatment times, all measurements for horses with left-sided lameness at time 1 (negative values) were inverted by multiplying them by −1. Changes in lameness amplitude were then calculated as the difference between Day 0 and subsequent time points (Days 1, 21, and 42). Given the differing threshold estimates for VS (8.5 mm for forelimb lameness) and PDmin and PDmax (3 mm each for hind limb lameness), lameness amplitude was standardized by dividing VS by 8.5 mm, PDmin and PDmax (if one was above threshold) by 3 mm, and the sum of PDmin and PDmax (if both were above threshold) by 6 mm. Trials in which lameness switched sides during subsequent time points were assigned a lameness amplitude of 0. A positive change in lameness amplitude indicated worsening, while a negative change indicated improvement.

A statistical analysis of changes in standardized lameness amplitude was performed to evaluate the effects of two treatments (“green/naMSC” and “red/aMSC”) across four time points (Days 0, 1, 21, and 42), with Day 0 serving as the control or baseline. The normality of the data (treatment x time x activity) was assessed using the Shapiro–Wilk test, which identified significant deviations from normality in several subgroups. Consequently, non-parametric statistical methods were employed. Within-group comparisons across time points for each treatment were analyzed using the Wilcoxon signed-rank test, and between-group comparisons at each time point were evaluated using the Mann–Whitney U test. The activities analyzed included straight movement, lunging to the left, and lunging to the right. A change in lameness amplitude between groups was considered statistically significant if *p* < 0.05. All analyses were conducted using an R script in the R base package within the RStudio development environment.

The sample size was initially established based on previous work by the co-authors. Upon completion of this study, a post hoc power analysis was performed (https://clincalc.com/stats/samplesize.aspx, accessed on 4 October 2024). The subjective lameness data for Day 0 and Day 42 from the aMSC-treated horses was used in the analysis and resulted in 90% power and an alpha of 0.05, thus suggesting that this study was adequately powered to determine a difference from Day 0 to Day 42.

## 3. Results

### 3.1. Horses

Horses were between the ages of 4 and 19 years old, with the average age being 12.75 years old. There were 11 mares and 13 geldings. Breeds included American Paint Horse, Appendix Quarter Horse, Hanoverian, Holsteiner, Irish Sport Horse, Missouri Fox Trotter, National Show Horse, Quarter Horse, thoroughbred, and mixed breed horse. The metacarpophalangeal joint was treated in 11 horses (six right, five left). The metatarsophalangeal joint was treated in seven horses (three right, four left). The middle carpal joint was treated in three horses (one right, two left). The radial carpal joint was treated in three horses (two right, one left).

Six horses were enrolled through Clinic #1, four horses were enrolled through Clinic #2, and fourteen horses were enrolled through Clinic #3. The median pre-study lameness score, as reported by the referring veterinarian, for all horses was 2.25 out of 5 (range 1.0–3.5). The median pre-study lameness score for aMSC-treated horses was 2.25 out of 5 (range 1.0–3.5). The median pre-study lameness score for naMSC-treated horses was 2.38 out of 5 (range 1.0–3.5).

### 3.2. Physical Examinations

Physical examinations, including rectal temperature, heart rate, and respiration rate, were performed at each lameness examination time point (Day 0, Day 1, Day 21, and Day 42). Vitals were considered “within normal limits” if the rectal temperature was less than 101.5°F, the pulse was equal to or below 48 beats per minute, and the respiration rate was less than or equal to 24 breaths per minute. At no point did any of the study horses have a rectal temperature outside the normal range. Elevated heart rates were reported four times in three horses. A description was provided on the information sheet that stated that the horses were “nervous” or “excited”. Elevated respiratory rates were reported three times with two horses. None of the increases in heart rate or respiratory rate occurred on Day 1. The authors found the elevations to be inconsistent across horses and treatments. The authors do not believe they are clinically relevant to this study.

Subcutaneous edema and increased swelling near the injection site were reported on Day 1 in three horses. All were aMSC-treated horses. Two were metacarpophalangeal joints, and one was a metatarsophalangeal joint. Two of these three horses received additional doses of flunixin meglumine for one and three additional days. Four different horses received additional doses of flunixin meglumine for one to two additional days based on the recommendations of the veterinarian. Subcutaneous edema and/or swelling were not listed for these cases, and the additional medication was recommended based on increased lameness. Three of these horses were aMSC-treated horses, and one was an naMSC-treated horse.

### 3.3. Subjective Lameness Scores

The mean subjective lameness grade for the straight line on Day 0 for naMSC-treated horses was 1.92 (±0.90), and for aMSC-treated horses, it was 2.04 (±0.94). There was no significant difference between these baseline values of aMSC- and naMSC-treated horses. Both the naMSC and aMSC treatments resulted in increased straight-line lameness on Day 1 (2.38 (±0.86) and 2.55 (±0.85), respectively), but they were neither significantly different than Day 0 nor each other. There was no significant difference between the treatments on Day 21 and Day 42. naMSC-treated horses had straight-line lameness scores that were significantly lower on Day 21 (1.0 ± 1.15) than on Day 1 (*p* < 0.001) and Day 0 (*p* = 0.0098). This trend continued, with Day 42 scores (1.13 ± 1.00) being significantly lower than Day 1 (*p* < 0.0001) and Day 0 (*p* = 0.0418) scores. aMSC-treated horses had straight-line lameness scores that were significantly lower on Day 21 (0.96 ± 1.03) than on Day 1 (*p* < 0.0001) and Day 0 (*p* = 0.0011). This trend continued, with Day 42 scores (0.79 ± 1.05) being significantly lower than Day 1 (*p* < 0.0001) and Day 0 (*p* < 0.0001) scores. When specifically evaluated from Day 0 to Day 42, the naMSC-treated horses improved by 0.79 of a lameness grade, and aMSC-treated horses improved by 1.25 lameness grades. The results for the subjective scores for the straight line, left and right circles, range of motion, flexion response, and effusion are listed in Table 1. Subjective lameness on the straight line is represented in Figure 1.

**Table 1 animals-14-03406-t001:** Summary statistics for subjective scoring of lameness parameters. Data are listed as mean (S.D). Different superscripts within the same treatment group and the same assessment parameter represent values that are significantly different (*p* < 0.05). Columns marked by an asterisk (*) represent data in which normality was not met, and the data were log-transformed for analysis. The summary statistics for these parameters are listed as median (interquartile range).

Treatment	Day	Straight Line (0–5) (Figure 1)	Circle Left (0–5)	Circle Right (0–5)	Flexion * (0–4) (Figure 2)	Range of Motion * (0–4) (Figure 3)	Effusion * (0–4) (Figure 4)
naMSC	0	1.92 (0.9) ^a^	2.04 (1.01) ^a,b^	2.17 (1.03) ^a^	3.00 (1.5–3.0) ^a^	2.00 (0.5–3.0) ^a^	3.00 (1.5–3.0) ^a,b^
	1	2.38 (0.86) ^a^	2.46 (1.01) ^a^	2.5 (1.07) ^a^	NP	NP	3.00 (2.0–3.0) ^a^
	21	1.00 (1.15) ^b^	1.00 (0.93) ^c^	1.13 (0.68) ^b^	1.50 (1.0–2.5) ^a,b^	1.00 (0.0–2.0) ^a^	2.50 (1.5–3.0) ^a,b^
	42	1.13 (1.00) ^b^	1.29 (0.89) ^b,c^	1.042 (0.62) ^b^	1.00 (0.0–1.0) ^a^	0.50 (0.0–1.0) ^a^	2.00 (0.5–3.0) ^b^
aMSC	0	2.04 (0.94) ^a^	1.71 (0.99) ^a,b^	2.00 (0.67) ^a,b^	2.00 (2.0–3.0) ^a^	2.00 (0.0–3.0) ^a^	2.00 (0.5–2.0) ^a^
	1	2.55 (0.85) ^a^	2.4 (0.74) ^a^	2.55 (1.07) ^a^	NP	NP	3.00 (2.0–3.0) ^b^
	21	0.96 (1.03) ^b^	0.83 (0.94) ^c^	1.21 (1.14) ^b^	0.50 (0.0–2.0) ^b^	0.00 (0.0–2.0) ^a^	2.00 (1.0–2.0) ^a^
	42	0.79 (1.05) ^b^	0.92 (1.06) ^b,c^	1.17 (1.27) ^b^	0.50 (0.0–2.5) ^b^	0.00 (0.0–1.0) ^a^	0.5 (0.0–2.0) ^a^

### 3.4. Subjective Assessments of Joint Effusion, Range of Motion, and Response to Flexion

Effusion scores of the target joint were significantly lower on Day 42 (2.00 [0.5–3.0]) than on Day 1 (3.00 [2.0–3.0], *p* = 0.0223) for horses treated with naMSCs. The median effusion score did not increase from Day 0 to Day 1 for naMSC-treated horses. For aMSC-treated horses, effusion scores were significantly higher on Day 1 (3.00 [2.0–3.0]) compared to Day 0 (2.00 [0.5–2.0], *p* = 0.0111), Day 21 (2.00 [1.0–2.0], *p* = 0.0111), and Day 42 (0.5 [0.0–2.0], *p* = 0.0111). Effusion scores are represented in Figure 4.

There were no statistically significant differences between time points or treatments for range of motion scores (Figure 3, Table 2). Scores assigned for response to flexion of the target joint were significantly lower on Day 42 (1.00 [0.0–1.0]) than Day 0 (3.00 [1.5–3.0], *p* = 0.0003) for naMSC-treated horses. For aMSC-treated horses, the response to flexion scores were significantly lower on Day 21 (0.50 [0.0–2.0], *p* = 0.0201) and Day 42 (0.50 [0.0–2.5], *p* = 0.0125) than on Day 0 (2.00 [2.0–3.0]). Response to flexion scores are represented in Figure 2.

### 3.5. Objective Lameness Assessment

When evaluating the straight-line objective lameness data, Day 21 lameness values were found to be significantly lower than those of Day 1 for naMSC-treated horses (*p* = 0.009) (Figure 5). No other timepoint comparisons were found to be significantly different. There was also no significance between naMSC-treated horses and aMSC-treated horses.

When evaluating the lunging to the left objective lameness data, the Day 1 lameness data values were significantly greater for both naMSC- and aMSC-treated horses compared to those of Day 0. For naMSC-treated horses, Day 21 lameness was significantly lower than Day 1 lameness (*p* = 0.041). For aMSC-treated horses, Day 1 lameness was significantly greater than Day 0 (*p* = 0.002) and Day 21 (*p* = 0.027) lameness. There was no statistical difference between naMSC- and aMSC-treated horses at any time point, nor was there a significant difference between Day 0 and Day 21 or Day 42.

When evaluating the lunging to the right objective lameness data, Day 1 lameness data values were significantly greater for horses treated with either naMSCs or aMSCs. For naMSC-treated horses, Day 42 lameness was significantly lower than Day 1 lameness (*p* = 0.032). For aMSC-treated horses, Day 1 lameness was significantly greater than Day 0 (*p* = 0.008) and Day 21 (*p* = 0.038) lameness. There was no statistical difference between naMSC- and aMSC-treated horses at any time point, nor was there a significant difference between Day 0 and Day 21 or Day 42.

### 3.6. Synovial Fluid Assessment

Synovial fluid was collected at the time of treatment injection from all but two horses. It was reported that no synovial fluid could be obtained on these two horses. None of the horses had a synovial fluid nucleated cell count that exceeded 2000 cells/uL. There was no significant difference between the nucleated cell counts for naMSC- and aMSC-treated horses (856.36 (±456.77) and 618.00 (±379.50), respectively). Half (11/22) of the synovial samples had total protein values greater than 2.0 g/dL. There was no significant difference between naMSC- and aMSC-treated horses (*p* = 0.778) (2.59 (±0.84) and 1.67 (±1.16) baseline synovial fluid total protein, respectively).

### 3.7. Owner Surveys

Fifteen (62.5%) Week 12 questionnaires were returned; seven were from aMSC-treated horses, and eight were from naMSC-treated horses. Two respondents (13.3%) reported that their horses had returned to a higher level of work compared to prior to the treatment injection. Eight respondents (53.3%) reported that their horses had returned to the same level of work compared to prior to the treatment injection. Three respondents (20%) reported that their horse had returned to a lower level of work compared to prior to the treatment injection. Two respondents (13.3%) reported that their horses had not returned to work. The summary of the performance levels reported by the owners is presented in Table 2. Only 1 of the 15 (6.7%) returned questionnaires reported an adverse effect, in which the owner indicated that due to the “improved condition of her carpus (treated joint), she is showing increased discomfort/lameness on the right (forelimb) and in the hocks”. This horse was in the naMSC group. All 15 owners who responded to the Week 12 questionnaire indicated that they were satisfied with their horses’ response to treatment. There was no statistical difference between aMSC- and naMSC-treated horses with regard to the reported level of work, adverse effects, or owner satisfaction in the 12-week owner questionnaires.

Twenty-three (95.8%) Week 18 questionnaires were returned; eleven were from aMSC-treated horses, and twelve were from naMSC-treated horses. Four respondents (17.4%) reported that their horses had returned to a higher level of work compared to prior to treatment injection. Eleven respondents (47.8%) reported that their horses had returned to the same level of work compared to prior to the treatment injection. Six respondents (26.1%) reported that their horses had returned to a lower level of work compared to prior to the treatment injection. Two respondents (8.7%) reported that their horses had not returned to work. The summary of the performance levels reported by the owners is presented in Table 2. Only 1 of the 23 (4.3%) returned questionnaires reported an adverse effect, in which the owner indicated that “she is visibly sore in her other joints due to the treatment in the injected one working, which I guess is a positive in a way”. The same owner reported an adverse effect in the Week 12 questionnaire. Eighteen (78.3%) responses indicated that owners were satisfied with their horses’ response to treatment. Five (21.7%) responses indicated that owners were not satisfied with their horses’ response to treatment, as follows: “Joint effusion is returning as is the lameness. Lameness is a lesser degree than before treatment, but the horse was sound at the time of the 12 week questionnaire”, “She still has a low grade lameness, we would have liked a full recovery. But we are satisfied with the treatment itself and believe it is and was her best avenue to return to full work”, “he was sound, however now that he is getting legged up, he is showing lameness again”, “don’t believe treatment worked. Returned to stall rest, did not respond how she hoped” and “Owner has run out of treatment options for their horse and is disappointed the treatment did not work”. Four of the five horses that had unsatisfied owners on the Week 18 questionnaire were aMSC-treated horses. There was no statistical difference between aMSC- and naMSC-treated horses for the reported level of work, adverse effects, or owner satisfaction in the 18-week owner questionnaires.

## 4. Discussion

In the present study, both intra-articular allogeneic MSC products, aMSC and naMSC, resulted in significant improvement in subjective lameness scores. When evaluating the subjective lameness grade on the straight line, 9 of 12 (75%) of the aMSC-treated horses and 8/12 (67%) of the naMSC-treated horses improved by one full lameness grade or more by 6 weeks following treatment. The average improvement in subjective lameness by Day 42 was 1.25 lameness grades for aMSC-treated horses. The only significant difference identified in the objective lameness data was between Day 1 and Day 21 for naMSC-treated horses. There was not a significant difference between the two products for any outcome parameter at any time point during the study period.

The use of mesenchymal stromal/stem cells (MSCs) for the treatment of orthopedic disease has been well described [8,9]. Early MSC applications touted their multipotent capabilities, stating that treatment goals involved healing damaged tissues with cells capable of differentiating into the target tissue. When MSCs are injected directly into the joint, they have been shown to populate the articular cartilage and synovium [10]. In time, the paracrine activity of MSCs and their specific cellular activity was recognized as their therapeutic role. The use of MSC therapy in humans continues to be highly debated [11,12,13]. Results have been repeatably positive in models of OA [2] and naturally occurring OA [14,15,16,17] in animals.

The advancement in our knowledge of MSCs has enabled us to prime or “activate” cells for specific purposes. Pezzanite et al. described toll-like receptor activation of MSCs to enhance antibacterial activity and immunomodulatory cytokine secretion [4]. Those authors tested the approach of using toll-like receptor-activated MSCs in an equine *S. aureus* septic arthritis model. They found that horses treated with intra-articular activated MSC therapy had significantly improved clinical signs compared to the horses that did not receive activated MSCs. The authors concluded that activated MSCs have improved antibacterial and cytokine responses in the equine septic joint model [5]. In this study, a proprietary method of activating MSCs was examined for their potential benefit in treating naturally occurring synovitis. The current study, however, did not find a statistical difference between activated and non-activated MSCs for treating synovitis. As described in our statistical section, we determined that this study was adequately powered for the evaluation of the progression from Day 0 to Day 42 within a treatment group. Further post hoc analysis determined that a sample size of over 130 horses per group would have been needed to determine a difference between the two treatment groups based on the subjective lameness scores. Due to the high number of horses required to find a statistical difference between treatment groups, the authors feel confident that there was no difference between the two treatments when evaluating lameness scores.

Based on our data, the authors feel that IA treatment with aMSCs or naMSCs is safe. There were no serious adverse events noted in this study. There was an increase in both subjective and objective lameness scores from Day 0 to Day 1. These increases were not significantly different from the baseline, nor were they related to one treatment group more than another. Seven (58.3%) aMSC-treated horses had subjective straight-line lameness scores that were increased on Day 1 compared to Day 0. Six (50.0%) naMSC-treated horses had increased subjective lameness scores on Day 1 versus Day 0. The term “flare” is often used to describe a 24- to 72-hour inflammatory response to an intra-articular injection. The term encompasses an increase in lameness and effusion, decreased range of motion, and increased local inflammation. Subcutaneous edema/swelling was reported in three (25.0%) aMSC-treated horses on Day 1. This would be consistent with a post-injection “flare”. The increase in effusion and lameness reported on Day 1 for both naMSCs and aMSCs would also be consistent with a “flare”. These adverse reactions are considered non-serious (minor) adverse events from a regulatory perspective. Furthermore, these events would be considered expected, as opposed to unexpected, since it is not uncommon for MSC therapies to result in short-lived local inflammation. Knott et al. reported, in a survey of veterinarians, that 83% of respondents had experienced local inflammation at the injection site following MSC treatments [18]. This is compared to 86% for platelet-rich plasma, 65.3% for autologous conditioned serum, and 55% for autologous protein solution [18]. A concern for increased immunological reactions from allogeneic MSCs in horses has been proposed, but this has had minimal evidential support [3]. Even the xenogeneic use of equine umbilical cord MSCs in dogs did not result in adverse reactions [15]. In the current study, the horses received a single dose of flunixin meglumine intravenously at the time of the MSC injection. With the potential for an increase in lameness scores on Day 1, a second dose of flunixin meglumine may be warranted. This may help with owner and veterinarian comfort, as post-injection flares can result.

The subjective lameness data resulted in more significant differences than the objective lameness data. This potentially could represent a subjective influence in the study design in that there was no negative control in the current study. The treating veterinarians were aware that the horses received some sort of treatment and may have been biased toward improvement and lower subjective scores. The patterns observed in the objective lameness data were similar to those in the subjective data. Donnell et al. reported that blinded subjective lameness grading and inertial sensor system had the greatest agreement in the carpal chip model of osteoarthritis [19], supporting that the combination of subjective assessment and inertial sensor systems provides the most consistent assessment. Due to the complexity of data produced by the inertial sensor system, an equally complex statistical analysis is required. The data must first be processed to calculate a difference from the baseline. Then, to assimilate the forelimb lamenesses and hindlimb lamenesses into a similar magnitude of lameness, the values are further transformed. The resulting summary statistics are based on data that have been transformed from the original data output. This can make it difficult to identify a statistically significant difference. A post hoc sample size assessment utilizing the final summary statistics for naMSC-treated horses Day 0 (1.96 ± 1.66) versus Day 42 (1.79 ± 1.99) revealed that nearly 1500 horses would be needed per group to achieve adequate power.

More owners of aMSC-treated horses reported not being satisfied in Week 18. Owner questionnaires are frequently utilized in clinical trials. They are valuable to monitor owner satisfaction, which is an important component of the willingness of clients to try new therapies. Questionnaires, however, are not without their challenges. Completion rates in the current study were 62.5% for the 12-week questionnaire and 95.8% for the 18-week questionnaire. This is quite good, as the average online questionnaire response rate is 44.1% [20], and one particular veterinary survey evaluating pain management in dogs and cats only reported a 4.4% response rate [21]. The current study offered an incentive for participation in the clinical trial but required that owners complete both questionnaires to receive the incentive. This likely aided in the higher-than-average questionnaire completion rate. Early participants did not complete the 12-week questionnaire, and the authors had to utilize emails and phone calls to accomplish questionnaire completion. Often, by the time we were successful, it was already Week 18, and therefore, we only obtained one questionnaire. This is important because it should be noted that no owners reported that they were not satisfied with the treatment in the Week 12 questionnaire, but five owners reported that they were not satisfied on the Week 18 questionnaire. Only one of those five had completed the 12-week questionnaire.

## 5. Conclusions

In conclusion, it was found that both activated and non-activated allogeneic equine umbilical cord blood-derived mesenchymal stromal cells resulted in significantly improved subjective lameness scores in horses with pain localized to the metacarpophalangeal, metatarsophalangeal, radial carpal, or middle carpal joints. The current study supports the reported beneficial effects of activated MSCs, with an overall improvement in subjective lameness of more than one lameness grade (1.25-grade improvement versus 0.79-grade improvement of naMSC-treated horses). Both treatments are a viable option for off-the-shelf allogeneic MSC joint treatments.

## Figures and Tables

**Figure 1 animals-14-03406-f001:**
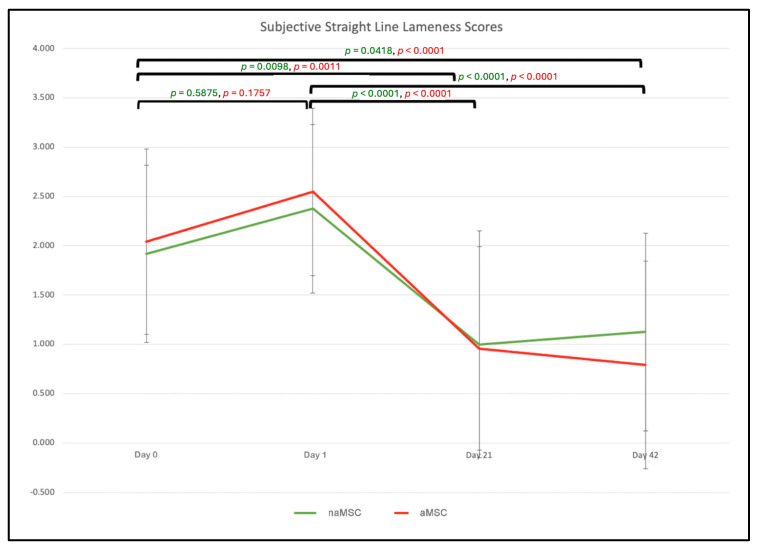
Mean subjective lameness scores for straight-line assessment. Lameness was graded on a 0–5 scale, which is described in Appendix A.

**Figure 2 animals-14-03406-f002:**
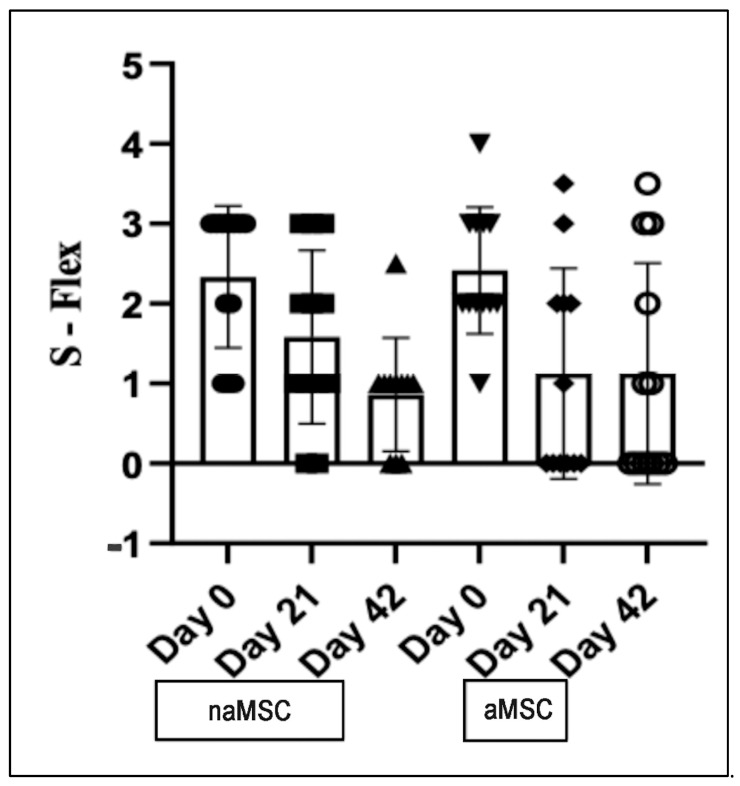
Median and interquartile range for subjective response to flexion of the target joint. Flexion was graded on a 0–4 scale, which is described in Appendix A. Data did not meet assumptions for normality and were log-transformed for analysis. Summary statistics for carpal flexion scores are represented with a box and whisker graph. Individual animals are represented by individual symbols.

**Figure 3 animals-14-03406-f003:**
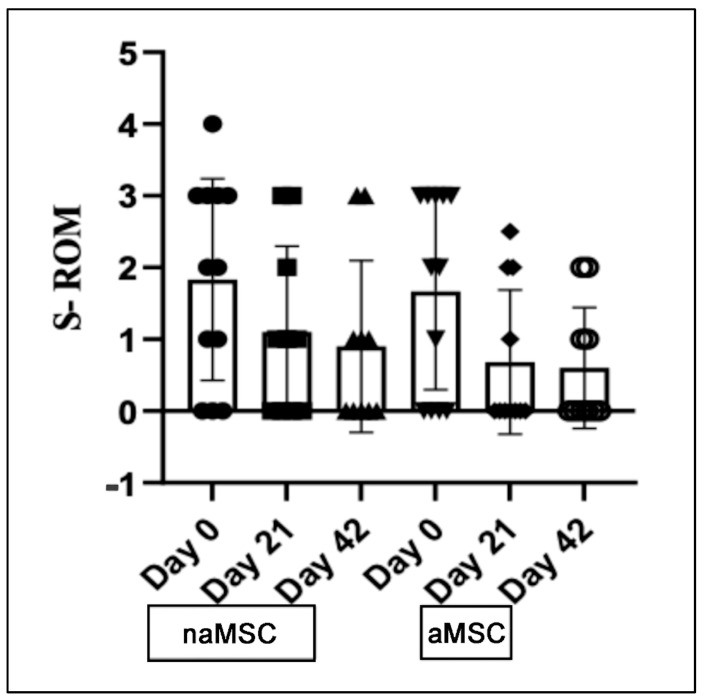
Median and interquartile range for subjective range of motion scores for the target joint. Range of motion was graded on a 0–4 scale, which is described in Appendix A. Data did not meet assumptions for normality and were log-transformed for analysis. Summary statistics for range of motion scores are represented with a box and whisker graph. Individual animals are represented by individual symbols.

**Figure 4 animals-14-03406-f004:**
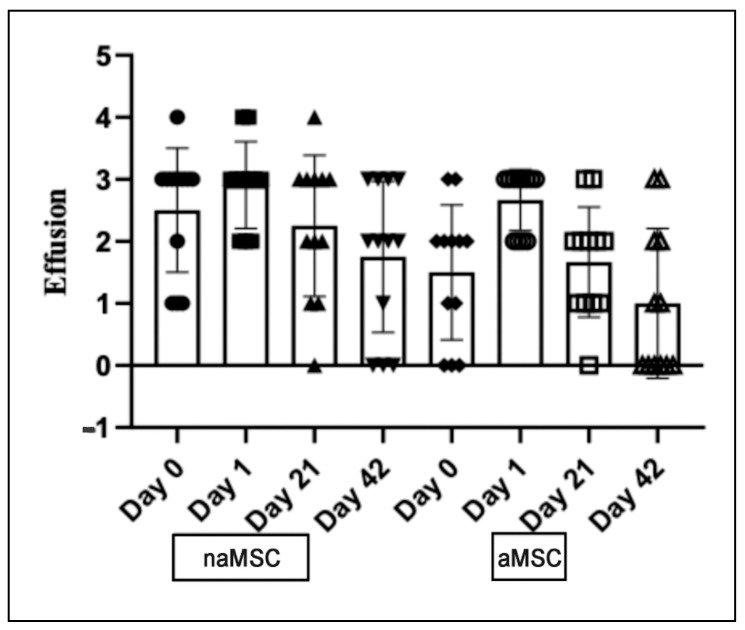
Median and interquartile range for subjective effusion scores of target joint. Effusion was graded on a 0–4 scale, which is described in Appendix A. Data did not meet assumptions for normality and were log-transformed for analysis. Summary statistics for effusion scores are represented with a box and whisker graph. Individual animals are represented by individual symbols.

**Figure 5 animals-14-03406-f005:**
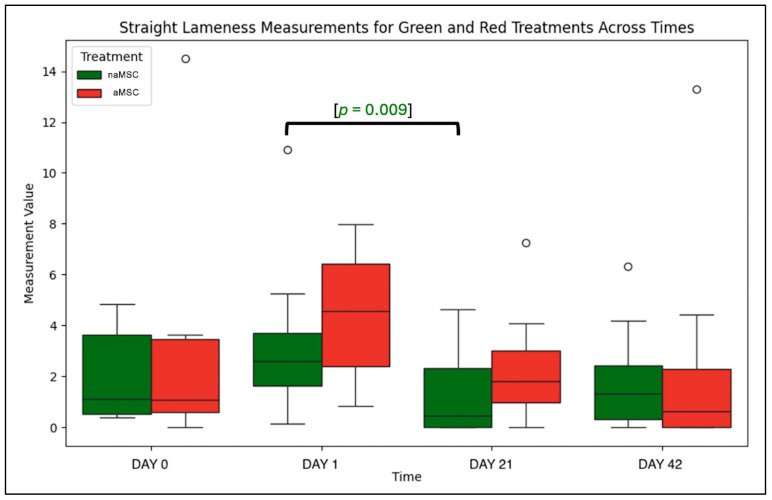
Objective lameness measurements on a straight line across time. naMSC-treated horses had significantly lower lameness scores on Day 21 than on Day 1 (*p* = 0.009). The circles represent outliers in the data set.

**Table 2 animals-14-03406-t002:** Results of reported performance level of horses in the Week 12 and Week 18 owner questionnaires.

Timepoint	Treatment	Performance Level				Questionnaires Returned
		Higher Level	Same Level	Lower Level	Has Not Returned to Work	
12 Weeks Post-Treatment	aMSC	1 (14.3%)	5 (71.4%)	1 (14.3%)	0 (0.0%)	7 (58.3%)
	naMSC	1 (12.5%)	3 (37.5%)	2 (25%)	2 (25%)	8 (66.7%)
18 Weeks Post-Treatment	aMSC	1 (9.1%)	5 (45.4%)	4 (36.4%)	1 (9.1%)	11 (91.7%)
	naMSC	3 (25.0%)	6 (50.0%)	2 (16.7%)	1 (8.3%)	12 (100%)

## Data Availability

The raw data supporting the conclusions of this article will be made available by the authors upon request.

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
