# Peer review of "A Pilot Study to Assess the Safety and Efficacy of Umbilical Cord Blood-Derived Mesenchymal Stromal Cells for the Treatment of Synovitis in Horses"

_animals, 2024, doi:10.3390/ani14233406_

Round 1
Reviewer 1 Report
Comments and Suggestions for Authors
Synovitis, a common joint condition in horses, can cause lameness and progress to osteoarthritis if untreated. This study evaluates the safety and effectiveness of a single injection of 10 million mesenchymal stromal cells (MSC), either activated or non-activated, in treating synovitis in 24 horses. Both MSC treatments led to notable lameness improvement, with no significant differences between the two methods, and owners reported high satisfaction with their horses' progress.
MAJOR CONCERNS-
The significant findings are scattered across long, detailed descriptions that make interpretation cumbersome. This could benefit from restructuring to prioritize and succinctly present key results. The section mixes various types of data (e.g., breed demographics, joint type treatments, and subjective vs. objective assessments) without clear transitions, making it challenging to follow.
Certain details, such as the specific heart rates of each horse on different days, are overly granular and detract from the main findings. These could be summarized more effectively.
Despite mentioning "significant" differences, it’s unclear whether all statistical comparisons are properly contextualized. For instance, the results mention a lack of significant difference between treatments for lameness scores but do not explore the clinical relevance of these findings, especially for practical applications.
Remove less relevant details (e.g., specific heart rates for each horse) to streamline the focus on major findings.
Lack of consistent baseline scores makes it difficult to attribute improvement solely to the treatment. Blinding is essential to avoid observer bias, especially when assessing subjective parameters.
Some adverse effects, such as subcutaneous edema and transient increases in heart rate, were noted but not analyzed in depth or compared across treatment groups.
Horses were evaluated only on Days 0, 1, 21, and 42, and some metrics (e.g., heart rate and respiratory rate) were recorded only sporadically. Limited and inconsistent timepoints miss key treatment effects, particularly short-term adverse reactions or delayed therapeutic effects
Comments on the Quality of English Languageenglish language is ok.
Author Response
Synovitis, a common joint condition in horses, can cause lameness and progress to osteoarthritis if untreated. This study evaluates the safety and effectiveness of a single injection of 10 million mesenchymal stromal cells (MSC), either activated or non-activated, in treating synovitis in 24 horses. Both MSC treatments led to notable lameness improvement, with no significant differences between the two methods, and owners reported high satisfaction with their horses' progress.
MAJOR CONCERNS-
The significant findings are scattered across long, detailed descriptions that make interpretation cumbersome. This could benefit from restructuring to prioritize and succinctly present key results. The section mixes various types of data (e.g., breed demographics, joint type treatments, and subjective vs. objective assessments) without clear transitions, making it challenging to follow.
Thank you for your comments. More clear distinctions have been made throughout the manuscript. Most notably, additional subsections have been added. We feel this significantly adds to the readability.
Certain details, such as the specific heart rates of each horse on different days, are overly granular and detract from the main findings. These could be summarized more effectively.
Thank you. This has been minimized. (Lines 228-236)
Despite mentioning "significant" differences, it’s unclear whether all statistical comparisons are properly contextualized. For instance, the results mention a lack of significant difference between treatments for lameness scores but do not explore the clinical relevance of these findings, especially for practical applications.
Thank you. We have made efforts through the results section and discussion to make “significant” differences more clear.
Remove less relevant details (e.g., specific heart rates for each horse) to streamline the focus on major findings.
Thank you this has been streamlined. (Lines 228-236)
Lack of consistent baseline scores makes it difficult to attribute improvement solely to the treatment. Blinding is essential to avoid observer bias, especially when assessing subjective parameters.
Thank you. We did collect baseline data. This was more clearly explained (Line 141) In regards to blinding, the participating veterinarians were blinded to whether the horse received naMSCs or aMSCs (Lines 99-100). But they did know that the horse had received some treatment.
Some adverse effects, such as subcutaneous edema and transient increases in heart rate, were noted but not analyzed in depth or compared across treatment groups.
Thank you. More descriptions and clarifications were added. (Lines 337-243)
Horses were evaluated only on Days 0, 1, 21, and 42, and some metrics (e.g., heart rate and respiratory rate) were recorded only sporadically. Limited and inconsistent timepoints miss key treatment effects, particularly short-term adverse reactions or delayed therapeutic effects
We appreciate your perspective here. Heart rate, respiratory rate and temperature were acquired at all 4 timepoints. Only elevations were noted and this has been clarified (Lines 231-236). The authors feel that there were consistent timepoints and that appropriate data was collected at each timepoint.
Reviewer 2 Report
Comments and Suggestions for Authors
The most significant deficiencies in this paper are the number of horses enrolled and the number of post-treatment surveys returned. However, I appreciate the difficulty of enrolling horses in a clinical trial and have good follow-through.
I suggest some changes to the manuscript to improve clarity.
Line 89: This should be Figure S3 (per the supplemental downloaded with the paper)
Line121-122: Suggest adding doses of flunixin in approximate mg/kg, and list types and approximate dose ranges in mg/kg
Line 136: Wording in parentheses at the end of the paragraph is unnecessary (redundant)
Line 144: This should be Figure S1 (per the supplemental downloaded with the paper)
Line 145" "to the owner" is unnecessary (redundant)
Line 151: This should be Figure S2 (per the supplemental downloaded with the paper)
Line 154: Suggest adding the information regarding incentives and contact with participants here (versus lines 357)
Lines 167-184: Suggest moving this out of the statistical analysis explanation. Suggest it should be with the evaluation/ examination methods (i.e., after Line 119)
Line 240: Should this be Figures S1-S4?
Line 273-274: Suggest removing subjective data provided by the owner on the survey and reporting only the objective data extracted from the survey. I suggest providing the subjective owner-reported data in a supplemental table.
Line 286-295: Suggest removing subjective data provided by the owner on the survey and reporting only the objective data extracted from the survey. I suggest providing the subjective owner-reported data in a supplemental table.
Line 325: Suggest adding "for the study period" or similar.
Line 332: Suggest defining "flares" or using different wording to describe
Author Response
Reviewer 2
The most significant deficiencies in this paper are the number of horses enrolled and the number of post-treatment surveys returned. However, I appreciate the difficulty of enrolling horses in a clinical trial and have good follow-through.
I suggest some changes to the manuscript to improve clarity.
Line 89: This should be Figure S3 (per the supplemental downloaded with the paper)
Thank you. We had supplemental figures out of order. What was listed in the manuscript is correct and the authors have made corrections to the supplemental files.
Line121-122: Suggest adding doses of flunixin in approximate mg/kg, and list types and approximate dose ranges in mg/kg
Thank you a dose range has been added (Line 142)
Line 136: Wording in parentheses at the end of the paragraph is unnecessary (redundant)
Thank you. This has been removed.
Line 144: This should be Figure S1 (per the supplemental downloaded with the paper)
Thank you. Our supplemental figures were out of order. What was listed in the manuscript is correct and we have made corrections to the supplemental files.
Line 145" "to the owner" is unnecessary (redundant)
Thank you. The redundancy of owner has been removed from this sentence.
Line 151: This should be Figure S2 (per the supplemental downloaded with the paper)
Thank you. I had my supplemental figures out of order. What was listed in the manuscript is correct and I have made corrections to the supplemental files.
Line 154: Suggest adding the information regarding incentives and contact with participants here (versus lines 357)
Thank you. This has been added to Materials and Methods (Lines166-168).
Lines 167-184: Suggest moving this out of the statistical analysis explanation. Suggest it should be with the evaluation/ examination methods (i.e., after Line 119)
Due to comment from the other reviewer stating that the sections were hard to follow and mixed various bits of information we have opted to leave this portion in the statistics but a sentence referring to the section was added (Lines 120-121).
Line 240: Should this be Figures S1-S4?
No these are included in the manuscript so are Figures 1-4 (not supplemental)
Line 273-274: Suggest removing subjective data provided by the owner on the survey and reporting only the objective data extracted from the survey. I suggest providing the subjective owner-reported data in a supplemental table.
Subjective and objective data is referring to the clinical lameness exams performed by the veterinarians. This has been clarified in the M&M (Lines 118 -120). All data from the owner questionnaires is subjective.
Line 286-295: Suggest removing subjective data provided by the owner on the survey and reporting only the objective data extracted from the survey. I suggest providing the subjective owner-reported data in a supplemental table.
Subjective and objective data is referring to the clinical lameness exams performed by the veterinarians. This has been clarified in the M&M (Lines 118 -120). All data from the owner questionnaires is subjective. Table 2 is the only table regarding the results from the owner questionnaires. We defer to the editor regarding whether it should be in the supplemental section.
Line 325: Suggest adding "for the study period" or similar.
We have added “during the study period.” (Line 364)
Line 332: Suggest defining "flares" or using different wording to describe
Great point! This has been done and further elaborated upon (Lines 390-398)
Round 2
Reviewer 1 Report
Comments and Suggestions for Authors
The authors have reviewed and addressed all the comments and suggestions provided by the reviewers. Based on their revisions and the improvements made to the manuscript, the manuscript can now be considered accepted for publication.